# Mechanical Properties of AISI 316L Lattice Structures via Laser Powder Bed Fusion as a Function of Unit Cell Features

**DOI:** 10.3390/ma16031025

**Published:** 2023-01-23

**Authors:** Luis H. Olivas-Alanis, Antonio Abraham Fraga-Martínez, Erika García-López, Omar Lopez-Botello, Elisa Vazquez-Lepe, Enrique Cuan-Urquizo, Ciro A. Rodriguez

**Affiliations:** 1Tecnologico de Monterrey, School of Engineering and Sciences, Av. Eugenio Garza Sada 2501, Monterrey 64849, Mexico; 2Laboratorio Nacional de Manufactura Aditiva y Digital MADiT, Autopista al Aeropuerto, Km. 9.5, Calle Alianza Norte 100, Parque PIIT, Apodaca 66629, Mexico; 3Tecnologico de Monterrey, School of Engineering and Sciences, Epigmenio González 500, Querétaro 76130, Mexico; 4Tecnologico de Monterrey, Institute of Advanced Materials for Sustainable Manufacturing, Av. Eugenio Garza Sada 2501, Monterrey 64849, Mexico

**Keywords:** additive manufacturing, laser powder bed fusion, cellular materials, stiffness tailoring

## Abstract

The growth of additive manufacturing processes has enabled the production of complex and smart structures. These fabrication techniques have led research efforts to focus on the application of cellular materials, which are known for their thermal and mechanical benefits. Herein, we studied the mechanical behavior of stainless-steel (AISI 316L) lattice structures both experimentally and computationally. The lattice architectures were body-centered cubic, hexagonal vertex centroid, and tetrahedron in two cell sizes and at two different rotation angles. A preliminary computational study assessed the deformation behavior of porous cylindrical samples under compression. After the simulation results, selected samples were manufactured via laser powder bed fusion. The results showed the effects of the pore architecture, unit cell size, and orientation on the reduction in the mechanical properties. The relative densities between 23% and 69% showed a decrease in the bulk material stiffness up to 93%. Furthermore, the different rotation angles resulted in a similar porosity level but different stiffnesses. The simulation analysis and experimental results indicate that the variation in the strut position with respect to the force affected the deformation mechanism. The tetrahedron unit cell showed the smallest variation in the elastic modulus and off-axis displacements due to the cell orientation. This study collected computational and experimental data for tuning the mechanical properties of lattice structures by changing the geometry, size, and orientation of the unit cell.

## 1. Introduction

Novel and promising additive manufacturing (AM) techniques have changed the design and production paradigm, enabling the fabrication of smart and complex structures that conventional manufacturing processes (i.e., casting, extrusion, and powder metallurgy) cannot achieve [1,2]. For metals processing, laser powder bed fusion (LPBF) is one of the most common methods thanks to the short production time, reduced economic impact, and low-volume production [3,4]. The combination of several process parameters (such as laser energy, laser scanning, and chamber environment) will influence the temperature and residual stress distribution, and hence, on the final part’s properties [5]. Even though this process accomplishes high-dense and near-net-shaping structures by iteratively laser melting metallic powder in a layer-by-layer process [6], the obtained surface quality is poor and rough compared with conventional machining [7]. This kind of surface finish will promote early failure due to the fact of stress concentrators and crack initiators encountered on the surface. Therefore, the application of postprocessing techniques, such as machining [8], chemical polishing [9], or abrasive finishing [10], are needed to enhance the final geometry and mechanical performance of the produced parts.

Several industries, such as aerospace, automotive, and medical, have aimed their efforts on tuning the structural properties by controlling the amount of material where it is needed through computer-aided design (CAD) [11]. The main approach focuses on the use of three-dimensional periodic open-porous cellular structures, commonly known as lattice materials or metamaterials [12]. Their application has shown promising mechanical results in light weighting [13], shock absorption [14], and stiffness matching [15]. Moreover, superior electromagnetic, acoustic, and thermal properties than those found in nature or on the bulk material have been observed on the architected materials [12,16].

The behavior of different metallic open-cell topologies under loading conditions has been studied for different applications. Among the most commonly employed until cells are the body-centered cube (BCC), face-centered cube (FCC), and simple cube (SC). According to Gibson and Ashby, the mechanical performance of the metamaterials is related to the level of porosity which, in turn, is associated with the pore architecture and the strut dimensions [17]. Furthermore, there is a threshold in the number of cell repetitions in the loading direction, where above this value, the structure’s stiffness will not be affected [18]. Hence, variations in any of these geometrical features will influence the mechanical response.

Changing the strut thickness is one of the most popular approaches. Previous research groups have achieved an elastic modulus as low as 0.02 GPa on LPBF-produced stainless-steel (AISI316L) BCC porous structures with a porosity level of 97% [19,20]. Similar studies by Serena Graziosi et al. [18] also showed the stiffness reduction by incrementing the interconnected porosity in BCC structures via variation of the strut thickness. Yang et al. varied the thickness of a gyroid structure for obtaining novel deformation patterns and mechanical properties. Additional works have explored this effect on other unit cell architectures, such as diamond [21], dodecahedron [22], and octahedron [23] manufactured via LPBF. The results showed the ability to tune mechanical properties by changes in the geometry. 

On the other hand, other works have observed alterations in the mechanical performance as a function of the strut angle and cell pose (orientation), although it has not been widely reported. Tsai et al. tested distinct strut angles on the same unit cell orientation, noticing a higher stiffness with an increasing strut angle [24]. Choy et al. observed a change in the deformation manner between two different unit cell orientations (0° and 90°) in SC and hexagonal honeycomb structures [25]. Similar research works resulted in improved mechanical properties by varying the unit cell orientation [26,27]. 

In order to enhance the understanding of lattice metamaterials, herein, we report the effect of the unit cell size and the cell rotation for tuning the compressive mechanical properties of three interconnected pore architectures: body-centered cube, hexagonal prism vertex centroid, and tetrahedron. A preliminary computational analysis was conducted to parametrically characterize the effect of the unit cell orientation. Next, experimental testing under compression was conducted on the selected AISI 316L porous samples produced by LPBF. 

## 2. Materials and Methods

### 2.1. Design of the Lattice Structures and Samples for Compression Test

The mechanical behavior under compression of three different unit cells with variation in the cell size and the cell orientation was studied. Figure 1 shows the three cell topologies assessed: body-centered cube (C), hexagonal prism vertex centroid (H), and tetrahedron (T). A body-centered cube consists of diagonal struts connecting each of the vertexes of the cube (Figure 1a). In the hexagonal prism vertex centroid, the struts also connect diagonally the vertexes of the body but within a hexagonal prism (Figure 1b). A tetrahedron’s struts follow the edges of the tetrahedral body (Figure 1c). The unit cell sizes of 3 × 3 × 3 mm and 6 × 6 × 6 mm were selected with a nominal strut thickness (*t*) of 800 μm for all of the constructs. The cell orientation with respect to the x- and y-axes was varied from 0° to 90° in 15° steps, as shown in Figure 1. 

The designed porosity was imposed on the cylindrical compression samples with an external diameter of 15 mm and a height of 30 mm, according to ISO 13314 standard [28], using Element software (nTopology, New York, NY, USA). Each sample was identified by the topology labels C, H, and T, and the cell size and the orientation angle were used as subindexes (i.e., C_3,45_).

### 2.2. Mechanical Computational Characterization via Finite Element Modeling (FEM)

The elastic property of the lattice structures (C, H, and T) was simulated in order to parametrically analyze the effect of the cell orientation under compressive loading. A computation analysis was performed on the structures with a 3 mm unit cell to reduce the size and edge effects [17,29]. The simulation analysis was conducted in the finite element modeling (FEM) module of the software nTopology (nTopology, NY, USA), as shown in Figure 2. nTopology allows for the numerical characterization of mechanical properties through meshing, preprocessing, and solving FE models. To achieve this, solid plates were added on the top and bottom of the lattice cylinders to ensure a load distribution over the effective area (〈*A*〉). The volumetric mesh and boundary conditions were created within the software. Tetrahedral elements with an edge length (element size) of 0.1 mm were employed for the analysis. The models were fed with linear elastic AISI 316L material. This material was available from the software library, with an elastic modulus of *E* = 193 GPa and a Poisson ratio of *υ* = 0.28. The bottom face of the bottom plate was restrained in the 6 degrees of freedom. Finally, a load *F* of 1N was applied on the top face of the upper plate. 

The resulting displacement along the z-axis was used to compute, first, the apparent elastic modulus (〈*E*〉 = *F/*〈*A*〉) and, second, the normalized apparent elastic modulus (〈*E*〉*/E*_SS316_) for each sample to observe the stiffness variation. Based on these computational results, the cell orientation with the upper and lower mechanical properties of each topology was selected for manufacturing. Additionally, the magnitude of the x,y-plane structural displacement, referred to as off-axis displacement, was calculated. 

### 2.3. Additive Manufacturing via the Laser Powder Bed Fusion (LPBF) Process

The AISI 316L metallic powder was obtained from Renishaw (Renishaw, Wotton-under-Edge, England). Using scanning electron microscopy (SEM) EVOMA25 (Carl Zeiss, Oberkochen, Germany), the average measured particle size (*n* = 55) was 28.93 ± 7.70 μm (*D_10_*: 20.13 μm, *D_90_*: 37.56 μm). The chemical composition, provided by the supplier, is presented in Table 1.

The LPBF fabrication was conducted in a Renishaw AM400 machine equipped with a ytterbium fiber laser with 400 W of maximum power. The processing parameters, based on a previously published work [30], are shown in Table 2. The cylindrical samples were printed vertically, with the z-axis in Figure 1 and Figure 2 normal to the printing bed. Then, they were removed from the building platform by a wire electrical discharge machine (EDM). Finally, the samples were cleaned with bidistilled water in an ultrasonic bath for 30 min to remove the unmelted and loose powder.

### 2.4. Morphological Characterization

Figure 3 illustrates the locations used for measuring the strut thickness, *t*, of the manufactured samples with the use of a focus variation microscope, InfiniteFocus G5 (Bruker Alicona, Austria). The strut thickness was measured perpendicular to the strut’s axial direction. A total of eight measurements were taken for each sample: four on the top face (*t_T_)* and four on the lateral face *(t_L_)*. 

In order to obtain the average apparent experimental density 〈*ρ_E_*〉 of each sample arrangement, all of the samples were weighed via the XPR250 weight scale (Mettler-Toledo, Greifensee, Switzerland) to compare the sample mass with an ideal dense part, considering the material density reported by the manufacturer (7.99 gr/cm^3^) and following Equation (1) [31]:(1)$$\langle \rho_{E}\rangle =\left(\frac{v_{i}}{v_{o}}\right)*100\%$$
where *v_i_* is the solid volume of the porous structure, and *v_o_* is the material volume of an ideal dense part (known as the apparent volume). The nominal engineering density 〈*ρ_N_*〉 was also calculated according to Equation (1) by diving the designed volume by the apparent volume.

### 2.5. Mechanical Characterization via Compression Tests

The manufactured SS316L cylindrical samples were subjected to compressive loading at a constant strain rate of 1 × 10^−3^ s^−1^ [28] in an AG-X Series universal machine (SHIMADZU, Kyoto, Japan). Figure 4 shows the setup for the mechanical testing. The load was applied at the top face of the sample and parallel to the build direction. Five samples for each lattice type were characterized. Furthermore, compressive bulk properties were obtained from a solid LPBF-produced sample. The apparent stress 〈*σ*〉 and the apparent strain 〈*ε*〉 were calculated based on the external sample dimensions. Afterward, the normalized engineering elastic modulus 〈*E*〉*/E_AISI 316L_* and normalized apparent yield stress 〈*σ_y_*〉*/σ_yAISI 316L_* were obtained from the resulting stress–strain curves.

## 3. Results

### 3.1. Computational Characterization via Finite Element Modeling

The impact of the unit cell orientation (3 mm unit cell size) on the elastic properties of the lattice structures was analyzed via computational simulation. The resulting 〈*E*〉 was computed by dividing the stress exerted by the z-direction strain data. The apparent elastic modulus was normalized to the bulk material to observe the relative elasticity reduction as a function of the orientation angle.

Figure 5 shows the range of the elastic properties that could be achieved by varying the orientation of the unit cell with respect to the principal axis of the cylindrical samples; this, in turn, exhibited the possible capacity to tailor the effective mechanical properties. The highest decrease in the mechanical elasticity was observed in the H unit cell structure, with an elastic modulus reduction of approximately 94% compared to the bulk material. The largest variation in the stiffness was seen in the C unit cell (5.6%), while the H and T unit cells presented similar stiffness range variations (4.8%). Moreover, a similar apparent elastic modulus was seen on the T_3,0_ and the T_3,90_ structures, as well as the C_3,0_ and C_3,90_. These results indicate an isotropy tendency, since the orientation of the unit cell was the same at both rotation ends. Regarding the hexagonal until cell, there was a difference between the stiffness of the H_3,0_ and H_3,90_ unit cell, where a 90° rotation turns the struts to a higher angle with respect to the X,Y-plane.

Additionally, it is important to consider the two-dimensional displacement, perpendicular to the loading direction, that the structure suffers under compression. Anisotropic structures will show displacement apart from the one along the loading direction. Hence, the maximum off-axis displacement was also analyzed. As seen in Figure 6, the structures suffered an off-axis deformation between 0 and 0.01 μm when a load of 1 N was applied on the top face. For the H and T topologies, the largest displacement was detected at a 15° rotation, while for the C cell, it was observed at 60°. The H_3,15_ design featured the highest off-axis value of all of the structures. This could be explained due to the presence of another deformation mechanism, different than bending or stretching, which is twisting. It forces the unit cell to rotate and push the surrounding cells in a specific direction. The same behavior was observed in other unit cells (i.e., H_3,30_ and C_3,60_). Depending on the application, the minimal off-axis displacement would be desired, as seen in most of the T cells (i.e., T_3,90_), where there was very small deformation from the loading axis.

### 3.2. Morphological Characterization

According to the FEA results, the lower and upper stiffness limits were found at the 0° and 45° cell orientations; hence, these two unit cell orientations were selected to be manufactured. Table 3 presents the experimental parameters employed of the printed parts, and Figure 7 shows the LPBF-produced samples. 

These lattice structure samples resulted with inaccuracies in the final geometries. Table 4 shows the nominal strut thickness, along with their corresponding experimental results (average of eight replications for each lattice type and orientation) and the absolute error. The average strut thickness in all of the samples showed a relative increase of 13.98% compared to the nominal value of 800 μm. The mean strut thickness on the top face was *t_T_* = 899.40 μm, while those in the lateral view showed a greater thickness of *t_T_* = 924.30 μm. The optical evaluation showed wider struts caused by the attachment of particles around the solid volume, possibly as a result of the enlarged melt pool.

Thus, with the increase in strut thickness, the bulk volume and the experimental apparent density increased as well (represented by the positive error values in Table 4), with absolute differences of up to 15.92%. The resulting experimental apparent densities (*ρ_E_*) were in the range of 23.54% to 68.17 %. Figure 8 also presents the rise in the apparent density, as all the values were above the designed value (dashed line).

### 3.3. Mechanical Characterization via Compression Tests

Figure 9 shows a representative compression stress–strain curve for the different samples tested. As expected, there was a considerable reduction in the mechanical properties between the bulk sample and the porous samples: the greater the unit cell, the greater the effect. Note the three characteristic deformation stages of the porous structures under compression: elastic zone, plateau zone, and densification zone. According to the architecture, cell size, and rotation, the beginning of the plateau and densification zones varied. Furthermore, the plateau zone also presented as smooth or wavy curve, depending on the architecture of the cellular structure. The stress drop peaks (see the serrated values of the stress) were attributed to the buckling of the strut elements, as well as the fracture of the layers. Furthermore, the plateau zone in the 3 mm cell size looked smoother than that of the 6 mm cell size. This was attributed to the breakage of the layers as the displacement increased. In regard to the cell orientation, the 45° rotation samples showed curves above the unrotated unit cell.

With the obtained data, the normalized apparent Young’s modulus 〈*E*〉 and apparent 0.2% yield stress 〈*σ_y_*〉 were computed and plotted in Figure 10 as a function of the apparent density for the different lattice structure geometries, cell size, and orientation. It can be observed that both apparent mechanical properties (i.e., elasticity and yield strength) decreased as the relative density decreased [32,33,34,35]. 

With the obtained data it was possible to fit the effect on the apparent mechanical properties as a function of the engineering density to a power law, described by Equations (2) and (3) [17]: (2)$$\frac{\langle E\rangle }{E_{SS316L}}=C_{E}{\;\rho \;}^{n_{E}},$$
(3)$$\frac{\langle \sigma_{y}\rangle }{\sigma_{y_{SS316L}}}=C_{y}{\;\rho \;}^{n_{y}},$$
where *C_E_* and *C_y_* are the consistency factors, and *n_E_* and *n_y_* are the exponents describing the structural deformation behavior. Table 5 shows the *C* and *N* coefficients for the apparent elastic modulus and yield strength of the different pore architectures and cell rotations. Furthermore, Figure 10 plots, in dashed lines, the scaling laws along the experimental data.

## 4. Discussion

This study aimed to assess the effect on the compressive mechanical properties by tailoring the three-dimensional geometry features. The results of the lattice structures (C, H, and T) showcase the mechanical advantages of producing quasi-solid porous parts by one of the AM technologies. 

To evaluate the quality of the manufacturing process, the resulting density, referred to as process density, was assessed on a solid cylinder. A microporosity ratio of ~2.97%, inherent to this manufacturing process, was detected. This result is similar to those reported by other studies based on the LPBF processing of AISI 316L [36]. The resulting intrinsic internal microporosity could be due to the fact of insufficient melting of the metal particles (process-induced pores). Another factor is the presence of trapped gases in the powder bed or melt pool, referred to as gas-induced pores [34]. Further process experiments are required to reduce the microporosity to a minimum level.

Despite these kinds of defects, the bulk compressive strength (experimentally estimated at 470 to 500 MPa) is consistent with that reported in similar studies [37]. The bulk elastic modulus is known to vary depending on the process parameters and microporosity [38]. The LPBF-produced lattice structures featured an apparent density in the range of 25% to 68%, reaching apparent elastic properties as low as 0.90 MPa, with a yield strength of 12.28 MPa. In comparison, previous research reported the LPBF of highly porous BCC AISI 316L structures with elastic values of 0.02 GPa, with a relative porosity of 96.5% [19] and 0.05 GPa in a 94.7% relative porosity profile [20]. However, the obtained samples did show larger dimensions than the nominal values.

The variation in the strut thickness was attributed to the LPBF-related defects, as reported in previous works. One of them was the “staircase effect”, caused by the layer discretization of the curves and inclined surfaces [39]. It had more impact on the struts with bigger inclinations, as the strut orientation varied in our designed samples. When evaluating the struts’ thickness, the resulting dimensions varied depending on the measurement region, finding wider thickness measurements from the lateral view than on top (*t_L_* > *t_T_*). Consequently, the geometrical accuracy was dependent on the angle between the strut and the building direction [27]. Additionally, the nonoptimal process parameter promoted the widening of the melt pool along the scan track, resulting in the attachment of partially melted particles, an effect known as “balling” [40,41], hence increasing the bulk volume as well. This was further observed in the lattice materials, where the surface-to-volume ratio increased with the increasing porosity. The results show that the increase in the apparent density was greater in the smaller cell size (13.55 ± 1.89%) than in the larger ones (7.14 ± 1.01%), finding absolute errors up to 15.92%. As more small cells fit in the cylindric volume, a greater amount of surplus material (denser structures than designed) will accumulate. On the other hand, rotating the unit cell did not affect significatively the density deviation. 

These dimensional discrepancies will have a direct effect on the mechanical response, since the deformation mechanisms depend on the geometrical features (i.e., length, strut thickness, an cross-sectional area). In a work by Alaña et al., the LPBF-produced structures with a diamond unit cell featured a significant change in the elastic anisotropy due to the fact of manufacturing imperfections [42].

Even though the alteration of mechanical properties in cellular structures is linked to the level of interconnected porosity, our experimental results (illustrated in Figure 10), as well as other related works [25,26,27], showed differences in stiffness at similar porosity levels. The characterization of the anisotropy behavior of the structures is of great importance, if homogenous directional deformation is desired, as the performance during application could be under multidirectional forces. According to the computational simulation results, the different unit cell structures changed their mechanical properties when the unit cell pose shifted around the x- and y-axes. Depending on the arrangement of the struts (i.e., unit cell architecture), the imposed porosity will affect the behavior under loading of the construction at a different level. The stiffness started to increase as the unit cell orientation changed from 0° to 45°, where the highest value was found. Then, the stiffness decreased in the 45° to 90° range. Though the elasticity at 0° and 90° was the same for the C and T cells, this is not true in the H unit cell. Table 6 shows the maximum angle between the strut and the plane normal to the loading direction (*θ*) at each x–y rotation. As seen, the maximum angle values had the same behavior as the normalized apparent properties (Figure 5). Hence, the difference at both rotation ends in the H unit cell, as well as the difference due to the cell rotation angles, is attributed to the strut arrangement. 

The main factor for the elastic shift as a function of the orientation angle is the dominating deformation mechanism. This fluctuates between bending dominated and stretch dominated [43], depending on the strut position regarding the plane normal to the force (plane: x,y). To analyze the deformation manner, the strut-based structures can be modeled as beams with an end (free end) subjected to a force, *F*, and the other end working as a fixed support, as shown in Figure 11 [44]. Additional assumptions of the model are an isotropic and homogenous material, the same stress–strain behavior under tensile and compressive loading, and the neglection of the strut torsion effect due to the loading [45]. Since the force direction (z-direction) was not perpendicular to the beam’s principal axis, the displacement of the strut was due to the stretching and bending components being coupled. Furthermore, a lateral motion (along the x-axis) will occur as well. Hence, the maximum displacement can be obtained by Hook’s law (Equation (4)) and Euler–Bernoulli (Equation (5)) cantilever beam theory.
(4)$$\delta_{u}=\frac{F_{u}L}{AE}=\frac{F\sin\theta L}{AE}$$
(5)$$\delta_{v}=\frac{F_{v}L^{3}}{3EI}=\frac{F\cos\theta L^{3}}{3EI}$$
where *F* is the applied external force along the z-direction, and *F_u_* and *F_v_* are the components of *F* along the strut axial (*u*) and transversal (*v*) direction*,* respectively; *L* is the length of the strut; *A* is the cross-sectional area normal to the force; *I* is the second moment of area; *h* is the unit cell size; and *δ_z_*, *δ_x_*, *δ_u_*, and *δ_v_* are the resulting displacements in the *z*, *x*, *u,* and *v* directions. Thus, the total deformation in the direction of the load (*δ_z_*) and perpendicular to it (*δ_x_*) is composed by the displacement due to the fact of bending and stretching, as seen in Equations (6) and (7), respectively.
(6)$$\delta_{z}=\delta_{u}\sin\theta +\delta_{v}\cos\theta$$
(7)$$\delta_{x}=\delta_{u}\cos\theta -\delta_{v}\sin\theta$$

From this model analysis, it can be concluded that for similar strut geometries (1) the strut angle plays an important role in the loading conditions, thus changing the participation of the stretch or bending of the struts and the amount of deformation along the loading direction; (2) the displacement due to the fact of bending (Equation (5)) will be larger than that due to the fact of stretching (Equation (4)), following the third power scaling law; (3) if the strut thickness is constant, the higher the strut length (i.e., bigger unit cell size), the larger the deformation and lower the structural strength; (4) uniaxial force will also promote the structure to deform in a different direction than the load, which could stimulate other deformation mechanisms such as twisting. 

Stretch-dominated structures are recognized for having higher stiffness and strength than those deformed by bending, the latter having better energy absorption properties [14,44]. In this matter, body-centered cube (C structure) and hexagonal vertex centroid (H structure) cells are known as bending-dominated materials [46]. The struts bend around the connecting nodes under an external compressive or tensile force. On the other hand, according to Maxwell’s criteria (*M*), used to evaluate the structural rigidity, the tetrahedron unit cell (T structure) is a stretch-dominated construct (*M* ≥ 0) [44]. The high strut connectivity makes the structure deform in a stretching manner when stress is exerted. 

There is not only expected uniaxial deformation parallel to the loading direction. Off-axis displacement was proven by computation analysis and modeled after Equation (7). As the struts also move perpendicular to the force, other deformation mechanisms can be present, which provoke rotation around the z-axis and the x–y motion of the cell, affecting the neighbor structures, and ending in structural off-axis deformation. While the major off-axis displacement values were found computationally on the C and H unit cells, with the latter performing the larger motion value at 15°, T showed minimal bidimensional movement for the different angles. This is related to the type of deformation and the strut interconnection. As stretch-dominated structures have a lower degree of freedom, the structure is less flexible and will deform mainly in the force direction. The minimal variation was also observed in the computational stiffness results, showing max-to-min (〈E〉max/〈E〉min) ratios of 1.5, 1.73, and 1.28 for the C, H, and T unit cells, respectively. 

The differences between the structures can be further observed and analyzed by studying the experimental strength-to-weight ratio (〈σy〉/w), as seen in Figure 12. The rotation of the cell produced an increase in the strength-to-weight ratio, which was attributed to the improvement in the yield stress, observed in the experimental results. Moreover, when the unit cell size was bigger, the ratio decreased. In this case, the main factor was the reduction of the engineered density of the structure. Finally, also seen in the simulation results, the T samples featured larger strength-to-weight ratios than the other two architectures; hence, this topology offers higher structural strength with less material.

The experimental and computational results are in concordance with the literature and mechanical models reported [20,44,45,46]. The T unit cell featured a higher elastic modulus than the other unit cells. The stress–strain diagram showed characteristic curves for each type of deformation mechanism, more perceptible on the 6 mm cells, where a larger displacement was present due to the longer struts. Consequently, the load oscillation in the plateau region, owing to strut fracture and layer failure, was more noticeable in the 6 mm cell size, as buckling and bending occurred on a major scale. Other works have conducted similar studies to assess this effect of the cell size on the effective properties [27,47]. Stretch-dominated structures (T cell) will exhibit an onset before the plateau zone, mainly caused by the plastic buckling of the struts, while bend-dominated have smoother plateau zones due to the soft deformation manner [44]. Moreover, the experiment’s apparent properties had the same behavior as the simulation for both unit cell sizes and both selected angle rotations. The only exception was the apparent elasticity of the T_6,45_, where the rotation of the cell produced a modest decrease in the property, which was even lower than C and H. This unexpected behavior can be attributed to the manufacturing defects, discussed previously, along with the microstructural arrangement [27]. The differences between the simulated and experimental normalized properties can also be attributed to these factors, since the computational model did not consider the manufacturing process, as well as the higher density after production.

The normalized apparent properties as a function of the porosity behavior were modeled using a power law, following the work of Ashby et al. on open porous structures [48]. Figure 13 compares the behavior of the C (with no rotation) cell structure found in this study with similar structures reported in other works [20,46]. Our results lie close above Ashby’s open-porous model and follow the behavior found in the literature. 

The *C* and *n* are descriptive of the structural deformation mechanism. A prefactor value lower than one can be attributed to the misalignment of the struts to the loading direction, structural defects, and bending or buckling of the elements [49]. While an exponent value between one and two denotes a coupling of the stretching- and bending-dominated deformation [50], differences between curves will indicate a change in the deformation manner, as described in previous works [20,50]. In this work, we found structural variations depending on the architecture of the unit cell and the rotation angle. 

## 5. Conclusions

The application of cellular metallic structures has a promising future in several fields for the solution of some structural challenges. The most important feature relies on the capability of tailoring the mechanical properties before fabrication. In this work, SS316L porous structures with different topological features (architecture, size, and rotation) were manufactured to study their performance under compressive loadings. The present study set the experimental and computational data useful for the model of expected properties, showcasing the following findings:
The LPBF process tended to generate a strut diameter larger than intended, resulting in larger relative densities compared to the nominal values. Moreover, these deviations were dependent on the component direction according to the printing direction;It was found that the angle between the strut and the plane normal to the loading directions had a direct impact on the structural stiffness by shifting the deformation from bending to stretching dominated;The LPBF parts with different types of lattice geometries and unit cell size reduced the SS316L bulk elastic modulus between 47% and 93% by imposing a porosity that resulted in an experimental density in the range of 23% to 69%;Off-axis displacement was also observed for all of the unit cell types at different levels, attributed to axial motion provoked by the external stress and the presence of twisting deformation manner. The H cell performed the greatest movement, while the T cell had almost none;The simulation results indicate that the tetrahedron unit cell had the smallest E_max_/E_min_ ratio and off-axis displacement due to the fact of its stretch-dominating deformation. Additionally, the steady mechanical behavior was further seen in the experimental results, featuring desirable elastic modulus despite the resulting dimensional variation.

Towards an understanding of the mechanical properties of AM lattices, the next steps will include the optimization of the manufacturing process parameters to achieve fully dense as-designed parts and the application of post-treatment processes for geometrical integrity. Next, the characterization of the produced parts should include a comprehensive study of the microstructure and the presence of process porosity. Moreover, different loading scenarios should be explored, potentially considering (i) bending and torsion, as they can reveal deformation mechanisms hidden under simple tensile or compressive loads, and (ii) cyclic loadings. 

## Figures and Tables

**Figure 1 materials-16-01025-f001:**
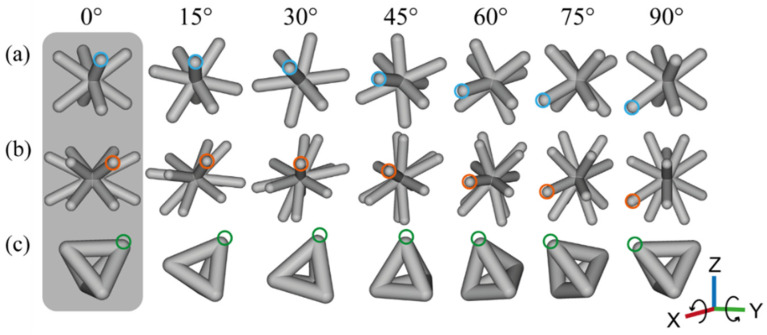
Unit cell architecture designs for the mechanical behavior assessment showing the resulting orientation as a function of the unit cell rotation from 0° to 90° in 15° steps: (**a**) body-centered cube (C) cell; (**b**) hexagonal prism vertex centroid (H) cell; (**c**) tetrahedron (T) cell. The circular marks are used to aid the visualization of the unit cell rotation.

**Figure 2 materials-16-01025-f002:**
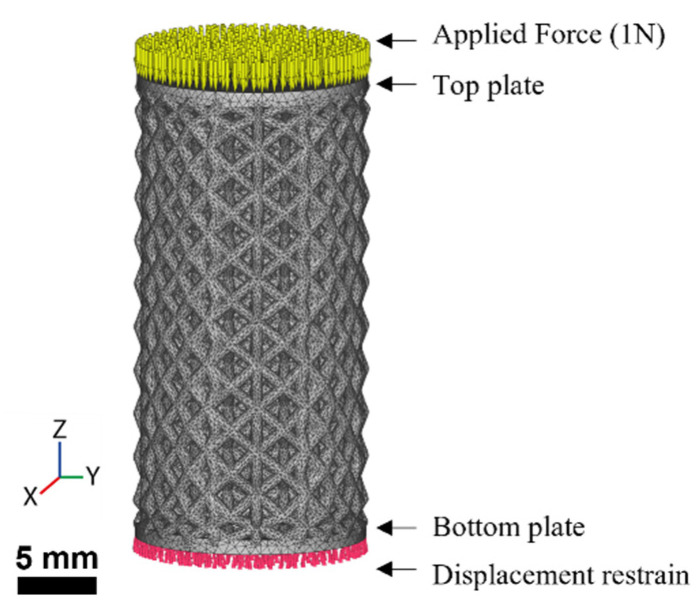
Finite element model (FEM) used for the computational characterization. Solid plates were added to the structure to distribute the applied force (yellow arrows) and the displacement restrain (red triangles) over the top and bottom surface, respectively. The resulting mesh is also shown in the structure.

**Figure 3 materials-16-01025-f003:**
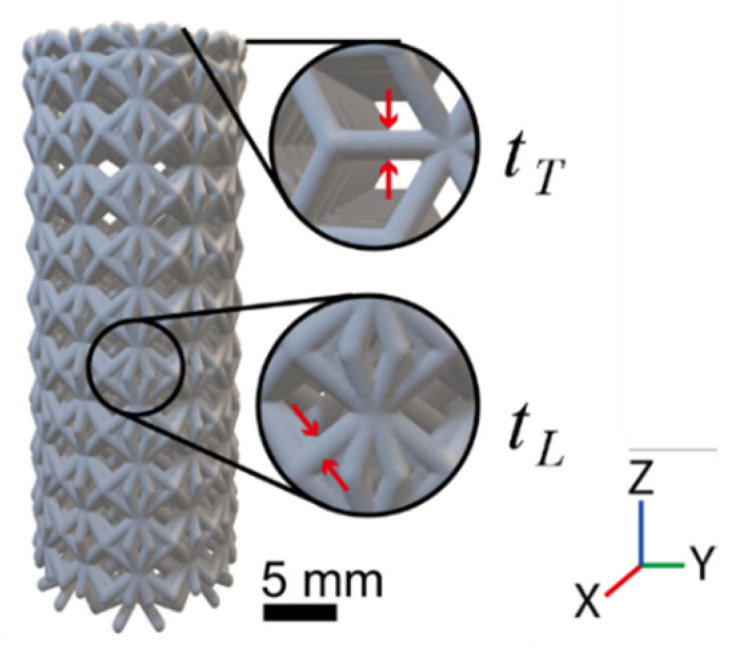
Strut thickness measuring locations for the porous structures. The measurements were taken at the top face (*t_T_*) and lateral side (*t_L_*).

**Figure 4 materials-16-01025-f004:**
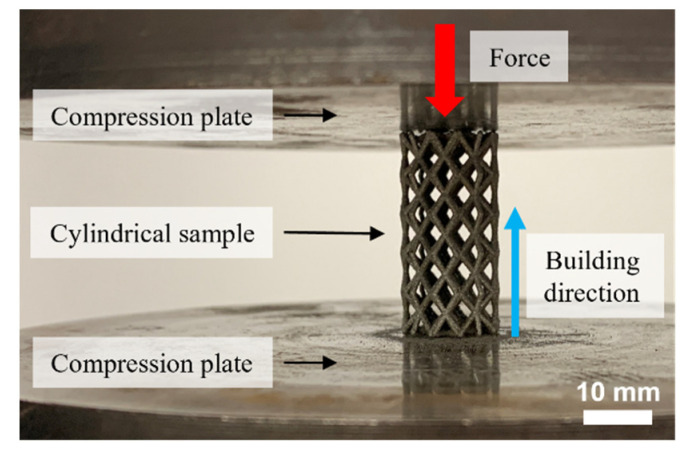
Mechanical characterization under the compressive loading setup. The samples were placed vertically and centered between two compression plates.

**Figure 5 materials-16-01025-f005:**
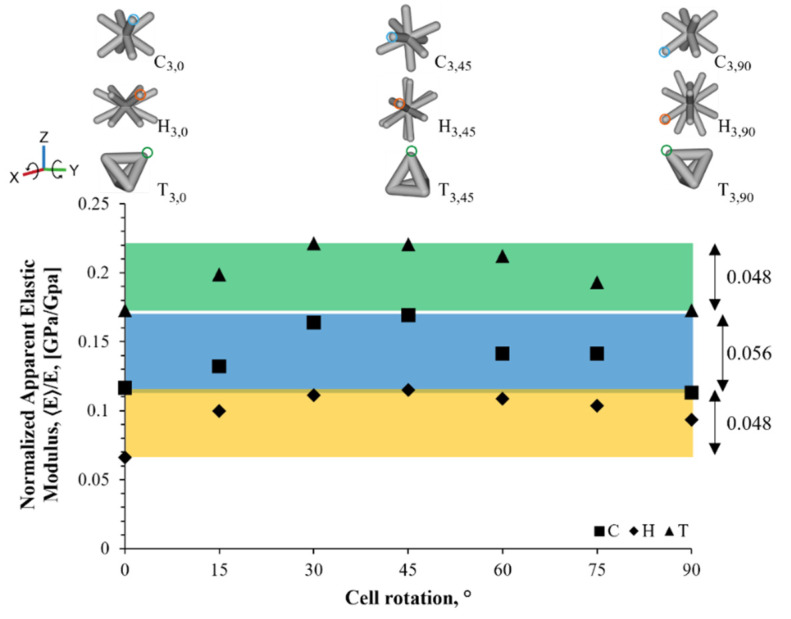
Normalized apparent elastic modulus variation plot as a function of the cell orientation for each unit cell geometry. The normalized apparent elastic modulus variation is shown as a range on the right-hand side of the plot.

**Figure 6 materials-16-01025-f006:**
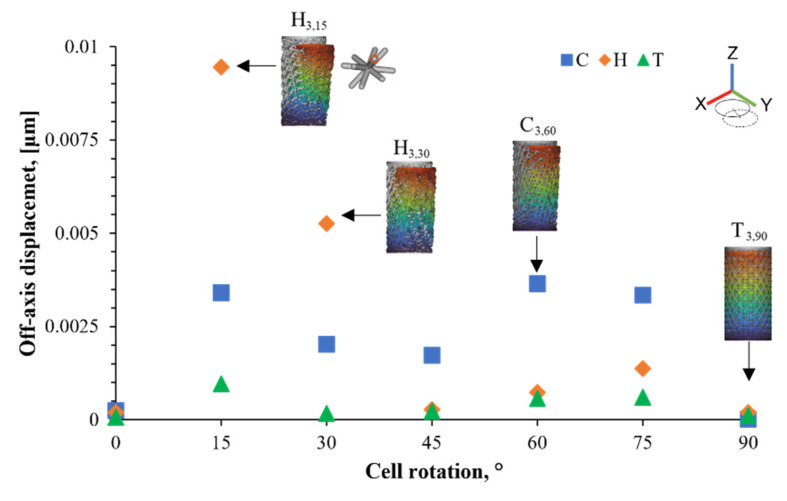
Off-axis displacement as a function of the cell orientation for each unit cell geometry. Undeformed structures are presented in grey, while deformed structures are shown in color (displacement was augmented by ×350,000). The unit cell with the most off-axis displacement (H3,15) is also shown.

**Figure 7 materials-16-01025-f007:**
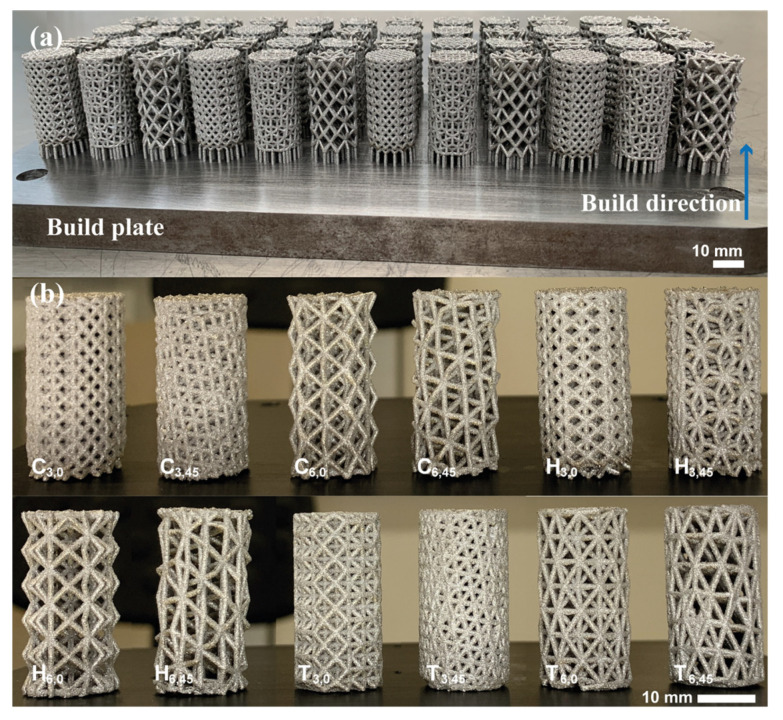
AISI 316L LPBF-produced porous cylinders for the compression mechanical testing: (**a**) samples after the LPBF process on the build plate; (**b**) samples removed from the build plate and cleaned.

**Figure 8 materials-16-01025-f008:**
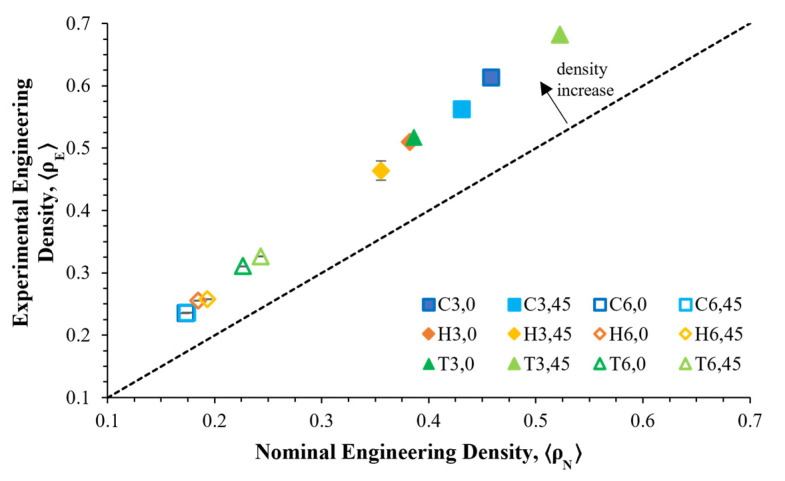
Comparison of the nominal engineering density (*ρ_N_*) and experimental engineering density (*ρ_E_*). The error bars show one standard deviation.

**Figure 9 materials-16-01025-f009:**
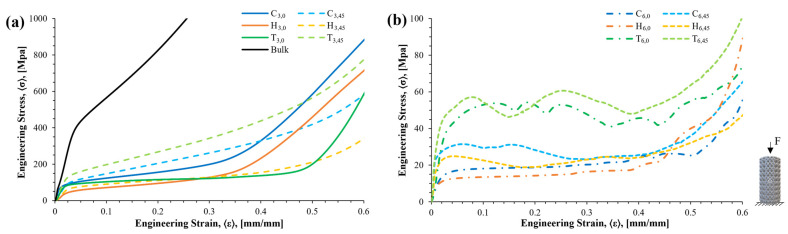
Engineering stress vs. engineering strain curves after compression testing: (**a**) bulk and 3 mm unit cell samples; (**b**) 6 mm unit cell samples.

**Figure 10 materials-16-01025-f010:**
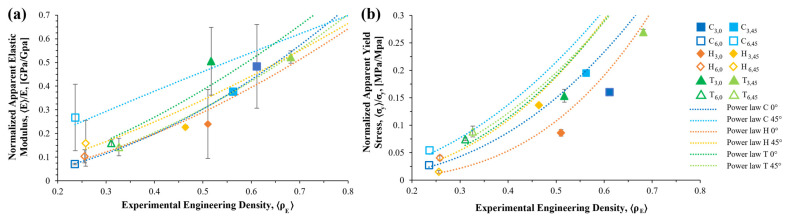
Normalized apparent mechanical properties vs. experimental engineering density 〈*ρ_E_*〉, along with the error bars, obtained from the compression stress–strain curves. Filled icons denote the 3 mm cell size, while the empty icons indicate the 6 mm cell size. Bold colors indicate a 0° and light colors a 45° cell orientation angle. (**a**) Engineering elastic modulus 〈*E*〉 plot; (**b**) engineering yield stress 〈*σ_y_*〉 plot.

**Figure 11 materials-16-01025-f011:**
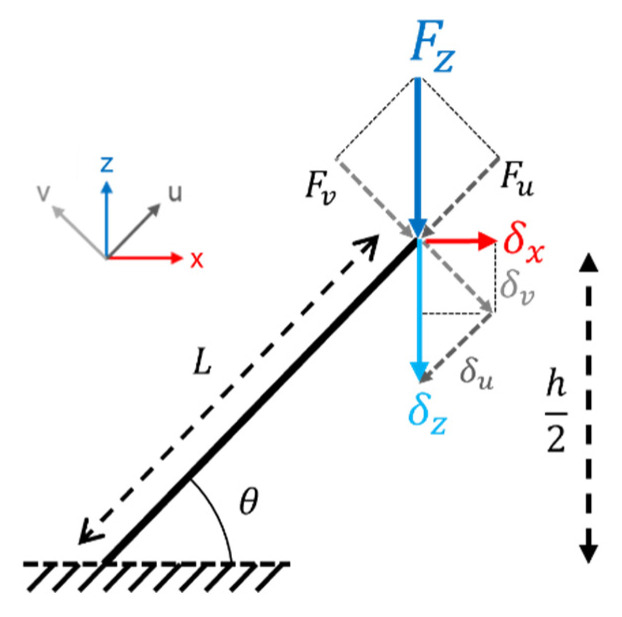
Diagram of the forces applied to a single strut in the unit cell. Equations (4) and (5) show the maximum displacement according to Hook´s law and Euler–Bernoulli’s theory for stretching and bending, respectively. *F_z_* is the applied external force along the z-direction, *F_u_* and *F_v_* are the strut’s axial (*u*) and transversal (*v*) components of *F_z_*, respectively; *L* is the length of the strut; *h* is the unit cell size; and *δ_z_*, *δ_x_*, *δ_u_*, and *δ_v_* are the resulting displacements in the z, x, u, and v directions.

**Figure 12 materials-16-01025-f012:**
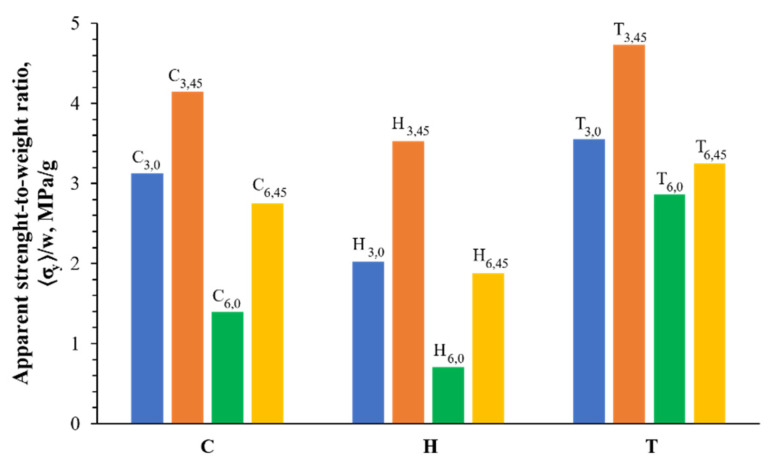
Apparent strength-to-weight ratios for the different configurations tested experimentally under compressive loading. The data are arranged by the geometry combination of the cell size and rotation.

**Figure 13 materials-16-01025-f013:**
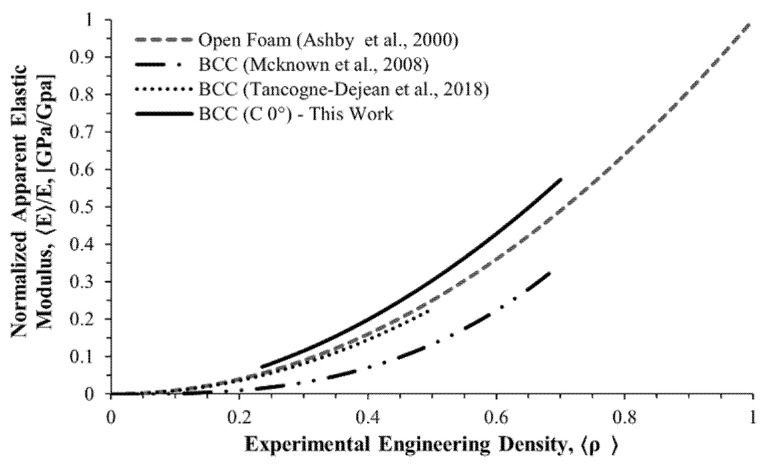
Normalized apparent elastic modulus as a function of the engineering density. The graph compares this study’s results with other reported works.

**Table 1 materials-16-01025-t001:** SS316L powder chemical composition.

Element	Fe	Cr	Ni	Mo	Mn	Si	N	O	P
Mass weight, (wt %)	Bal.	16.00–18.00	10.00–14.00	2.00–3.00	≤2.00	≤1.00	≤0.10	≤0.10	≤0.045

**Table 2 materials-16-01025-t002:** LPBF process parameters [30].

Parameter	Value
Laser power, *P*	170 W
Exposure time, *ET*	20 μs
Hatch distance, *HD*	60 μm
Layer thickness, *LT*	50 μm
Number of exposures, *N*	2
Point distance, *PD*	80 μm
Hatch offset, *HO*	60 μm
Strategy	Meander

**Table 3 materials-16-01025-t003:** Experimental parameters of the LPBF-produced samples for mechanical characterization.

ID	Cell Topology	Unit Cell Size	Cell Orientation
C_3,0_	Body-centered cube	3 mm	0°
C_3,45_			45
C_6,0_		6 mm	0°
C_6,45_			45
H_3,0_	Hexagonal prism vertex centroid	3 mm	0°
H_3,45_			45
H_6,0_		6 mm	0°
H_6,45_			45
T_3,0_	Tetrahedron	3 mm	0°
T_3,45_			45
T_6,0_		6 mm	0°
T_6,45_			45

**Table 4 materials-16-01025-t004:** Comparison of the nominal vs. experimental dimensions for the strut width.

ID	Nominal		Experimental			Absolute Error		
	Strut Thickness, *t*, (μm)	Engineering Density, 〈*ρ_N_*〉, (%)	Strut Top Thickness, *t_T_,* (μm)	Strut Lateral Thickness, *t_L_*, (μm)	Engineering Density, 〈*ρ_E_*〉, (%)	Strut Top Thickness, *t_T_, (*μm)	Strut Lateral Thickness, *t_L_,* (μm)	Engineering Density, 〈*ρ_E_*〉, (%)
C_3,0_	800	45.83	877.89	900.38	61.31	77.89	100.38	15.48
C_3,45_		43.09	868.69	880.16	56.25	68.69	80.16	13.16
C_6,0_		17.27	883.74	914.34	23.54	83.74	114.34	6.27
C_6,45_		17.42	881.42	896.11	23.62	81.42	96.11	6.2
H_3,0_		38.23	920.33	921.21	51.02	120.33	121.21	12.79
H_3,45_		35.52	882.13	909.65	46.39	82.13	109.65	10.87
H_6,0_		18.48	940.63	932.34	25.56	140.63	132.34	7.08
H_6,45_		19.29	897.51	945.82	25.79	97.51	145.82	6.50
T_3,0_		38.6	898.37	954.98	51.71	98.37	154.98	13.11
T_3,45_		52.25	910.70	945.65	68.17	110.7	145.65	15.92
T_6,0_		22.63	928.79	953.88	31.07	128.79	153.88	8.44
T_6,45_		24.3	902.61	946.90	32.64	102.61	146.9	8.34
Bulk		100.00			97.03			−2.97

**Table 5 materials-16-01025-t005:** Apparent properties power law prefactor (*C*), exponent (*n*), and correlation factor (*R^2^*).

ID	〈*E*〉/*E*_SS316L_			〈*σ_y_*〉/ *σ_y_*_SS316L_		
	*C* _E_	*n* _E_	*R* ^2^	*C* _y_	*n* _y_	*R* ^2^
C 0°	1.1227	1.8900	0.99	0.8097	2.4337	0.9702
C 45°	0.8506	0.8836	0.88	0.8647	2.0074	0.9773
H 0°	0.9372	1.6945	0.99	0.9469	3.135	0.9980
H 45°	0.9115	1.4150	0.98	1.0132	2.431	0.9991
T 0°	1.1586	1.5932	0.97	0.9416	2.3092	0.9925
T 45°	1.0471	1.7828	0.99	0.8489	2.1151	0.9945

**Table 6 materials-16-01025-t006:** Maximum strut angle to the plane normal to the loading direction.

	Unit Cell Rotation Angle						
ID	0°	15°	30°	45°	60°	75°	90°
C	35.26	56.35	76.33	80.26	63.42	47.79	35.26
H	26.57	46.61	63.43	67.99	62.42	63.37	63.44
T	45.00	57.43	62.11	58.60	66.72	59.29	45.00

## Data Availability

All data generated and analyzed in this work are included within the article.

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
