# Peer review of "Mechanical Properties of AISI 316L Lattice Structures via Laser Powder Bed Fusion as a Function of Unit Cell Features"

_materials, 2023, doi:10.3390/ma16031025_

Round 1
Reviewer 1 Report
The manuscript numbered "materials-2148029" was reviewed with great attention. The following comments are suggestions to improve paper quality:
The introduction needs some improvement.
Add more information about FEM software and modeling procedure.
Merge figures 2 and 4 and add a suitable scale bar for them.
Add a table related to the experimental parameters. It will have helped better understand IDs in table 3.
It seems repeatability studied, but there is no error bar for results.
Add SEM related to the fractography of the sample if it is possible.
Add strength to weight ratio diagram or table and discuss it.
There is an error related to bookmarks or hyperlinks in the text on page 4, line 136.
The following papers are suggested for the introduction section:
3D printing of bending-dominated soft lattices: numerical and experimental assessment
Modeling SEBM process of tantalum lattices
A comprehensive investigation of abrasive barrel finishing on hardness and manufacturability of laser-based powder bed fusion hollow components
Author Response
Document attached

Reviewer 2 Report
materials-2148029
“Mechanical properties of AISI 316L lattice structures via laser powder bed fusion as a function of unit cell features”studied the mechanical behavior of stainless steel (AISI 316L) lattice structures both experimentally and computationally. Results showed the effect of pore architecture, unit cell size, and orientation on the reduction of mechanical properties.
The manuscript after minor revision is suitable for publication in the journal of materials.
(1)Line136. There are some typos of references.
(2) Line 260, Figure 9b. The stress-strain curves of T6,0 and T6,45 have peaks. Why?
(3) Line376-Line380. The relations between the eq. (6) and eq. (7) and conclusions are not clear. Please rewrite the text.
Author Response
Document attached
